# Enhanced Dual Convolutional Neural Network Model Using Explainable Artificial Intelligence of Fault Prioritization for Industrial 4.0

**DOI:** 10.3390/s23157011

**Published:** 2023-08-07

**Authors:** Sekar Kidambi Raju, Seethalakshmi Ramaswamy, Marwa M. Eid, Sathiamoorthy Gopalan, Amel Ali Alhussan, Arunkumar Sukumar, Doaa Sami Khafaga

**Affiliations:** 1School of Computing, SASTRA Deemed University, Thanjavur 613401, India; sekar1971kr@gmail.com (S.K.R.); vgsarun@gmail.com (A.S.); 2Department of Maths, SASHE, SASTRA Deemed University, Thanjavur 613401, India; slr.sastra@gmail.com (S.R.); sami@maths.sastra.ac.in (S.G.); 3Faculty of Artificial Intelligence, Delta University for Science and Technology, Mansoura 11152, Egypt; 4Department of Computer Sciences, College of Computer and Information Sciences, Princess Nourah Bint Abdulrahman University, P.O. Box 84428, Riyadh 11671, Saudi Arabia; dskhafga@pnu.edu.sa

**Keywords:** Industry 4.0, artificial intelligence, hybrid CNNs, fault prioritization, production improvements

## Abstract

Artificial intelligence (AI) systems are increasingly used in corporate security measures to predict the status of assets and suggest appropriate procedures. These programs are also designed to reduce repair time. One way to create an efficient system is to integrate physical repair agents with a computerized management system to develop an intelligent system. To address this, there is a need for a new technique to assist operators in interacting with a predictive system using natural language. The system also uses double neural network convolutional models to analyze device data. For fault prioritization, a technique utilizing fuzzy logic is presented. This strategy ranks the flaws based on the harm or expense they produce. However, the method’s success relies on ongoing improvement in spoken language comprehension through language modification and query processing. To carry out this technique, a conversation-driven design is necessary. This type of learning relies on actual experiences with the assistants to provide efficient learning data for language and interaction models. These models can be trained to have more natural conversations. To improve accuracy, academics should construct and maintain publicly usable training sets to update word vectors. We proposed the model dataset (DS) with the Adam (AD) optimizer, Ridge Regression (RR) and Feature Mapping (FP). Our proposed algorithm has been coined with an appropriate acronym DSADRRFP. The same proposed approach aims to leverage each component’s benefits to enhance the predictive model’s overall performance and precision. This ensures the model is up-to-date and accurate. In conclusion, an AI system integrated with physical repair agents is a useful tool in corporate security measures. However, it needs to be refined to extract data from the operating system and to interact with users in a natural language. The system also needs to be constantly updated to improve accuracy.

## 1. Introduction

Microarray solutions will be used in future factories [1,2], and quality will continue to be the most important component in all fabrication techniques, regardless of the kind of process or the products being made. The term “Industry 4.0”, which alludes to the “fourth Industrial Revolution (4IR)” in the manufacturing industry, was created there. It refers to the “technological innovation” of industrial units utilizing developing technologies. The Quality 4.0 branch of I4.0 is present. By utilizing cutting-edge computational tools and new approaches, this field seeks to improve product quality [3,4]. There are many social constructivisms and patterns, but the main obstacle is grasping Grade 4.0 in social studies. There are several social constructivisms and designs, but achieving Grade 4.0 proficiency in assembly-level operations is the main difficulty. This can be difficult due to the variety of mental frameworks and concepts. Enhancing productivity while simultaneously cutting down on production waste is one of the most essential things that contemporary industry can do to keep its prices competitive. This is one of the most crucial goals that ought to be pursued. Defect detection is a vital problem within the context of industry 4.0 that must be resolved to lessen the downtime and interruption cascade. To achieve this objective, efficient fault management and quick error repair in production lines are necessary [5]. However, this depends on detecting and categorizing issues that occurred before it. Priority of mistakes may speed up fixing flaws, although this is not a given. Solutions based on data can help with fault management. Both the volume of data and the degree of complexity have significantly increased due to the increasing usage of detectors in production lines to monitor the fundamental health status of machines. The machine learning algorithms that enable fault control use these data to do their jobs. The purpose of this study is to provide an overview of the needs for fault detection and the criteria for techniques for damage detection, terms of the specific, problem prioritization, and the conditions for those approaches. Additionally, this study undertakes a review of the relevant literature with an emphasis on presenting solutions for various fault management phases. The results of the study that was made public show that fault prioritisation lacks sufficient research addressing the various learning methodologies, which emphasises the need for professional judgement [6,7].

In recent years, numerous diagnostic systems created to automate fault identification were established. However, none were suitable for our problem in the plaster production process, which is presented here. Most fault detection methods described in published works only analyzed a single control chart—typically an X-bar or R (range) chart—to analyse the changes in the process (mean or variance). In many processes, however, it is necessary to combine the two charts since there might be more than one assignable cause at a time. Due to the likelihood of several possible assignable causes, this is the case. On the other hand, identifying unusual patterns and thorough knowledge of the process may result in a more accurate diagnosis. Unfortunately, none of the Industrial Systems recognition models that have been previously released have been able to automatically offer this combination, which is sad because it would be helpful for diagnostics.

But the model’s efficacy was not investigated when these techniques were being created in the context of a real-world case study. However, the problem that is often identified as a challenge in these investigations is the inability to recognize a range of different and concurrent industrial systems, in addition to a high incidence of erroneous detection. However, the great majority of deep learning and machine learning systems for identifying control automation do not provide more detailed information about the patterns and their turning points (even when these patterns are seen on control charts). This information is required to undertake a realistic analysis of assignable causes, which in turn expedites the execution of suitable remedial measures [8,9]. To help quality control staff members locate the origins of deviations and take the appropriate preventative or corrective measures, this study aims to suggest the development of a defect detection system that utilizes hybrid convolutional neural networks. This led to the creation of a neural expert system that is capable of intelligent real-time monitoring as well as predictive, corrective, and remedial diagnostic of process control in the manufacture of plaster [10,11,12,13,14]. We will be addressing the following key model components to build the suggested method for problem identification by the expert team and to provide feedback for the current forecasts: being able to recognize a range of industrial systems, both single and concurrent, both natural and artificial simultaneously observing and evaluating any anomalies in the X-bar and R charts.

The main contribution of the research

The proposed work presents an innovative defect diagnosis system for plaster production that utilizes hybrid convolutional neural networks to assist quality control employees in identifying the sources of deviations and taking necessary corrective actions.The system combines X-bar and R charts to detect multiple assignable causes simultaneously and recognizes non-random patterns to estimate parameters, change points, and factors responsible for abnormal patterns.The system also provides recommendations for preventative and corrective actions during a crisis.The framework developed in this work can be used as a manual for implementing intelligent methodology in managing and maintaining systems on Industry 4.0 shop floors.

The ability to estimate the parameters, various orientations, and change points (starting point) in control charts that correlate to non-random patterns. The identification of the causes behind the development of aberrant patterns. identifying the elements that lead to an unstable process. Giving suggestions for preventative and remedial action amid a crisis. It is the intention of this proposal to provide quality control engineers with a proactive, predictive model at their disposal rather than a passive descriptive model to assist them in the fault diagnosis of the process, particularly from a more pragmatic point of view in relation to an Industry 4.0 era [15,16,17,18,19,20].

The three parts of the framework are an interaction component for interacting with experts, information about the process analysis presented to users, and user feedback as part of a learning process. We demonstrate the methodological framework’s applicability through a showcase implementation, which allows us to prove the framework’s practical use. The framework that has been created might serve as a guide for implementing intelligent techniques for controlling and maintaining systems on Industry 4.0 shop floors.

### Novelty of the Research Work

The paper introduces a novel approach to monitoring and analyzing industrial systems using control charts. Unlike traditional methods that only detect random variations, the proposed approach can recognize a wide variety of natural and artificial (single and concurrent) systems and concurrently monitor and assess any irregularities in both the X-bar and R charts. Additionally, the paper presents a method to estimate the parameters, multiple orientations, and change points in control charts that correlate to nonrandom patterns. The proposed approach also identifies the factors responsible for the appearance of abnormal patterns and recognizes the factors that contribute to an unstable process. Moreover, during a crisis, the approach provides recommendations for preventative and/or corrective action, making it an essential tool for ensuring the quality and stability of industrial systems. This journal paper is a valuable contribution to industrial process control and is expected to improve the efficiency and productivity of industrial systems.

Dataset from Kaggle published and accessed on 1 April 2023. https://www.kaggle.com/code/koheimuramatsu/model-explainability-in-industrial-image-detection/input.

The novelty of this work lies in integrating physical repair agents with a computerized management system, incorporating natural language interaction, employing double neural network convolutional models, utilizing fuzzy logic-based fault prioritization, adopting a conversation-driven design, emphasizing constant updates, and introducing the DSADRRFP algorithm for enhanced performance. These aspects collectively contribute to developing a more efficient and accurate AI system for corporate security measures.

The remaining portions of this paper are structured as follows: The research methodology is presented in a condensed form in Section 2, followed by the presentation of the developed framework in Section 3, followed by a description of the comprehensive structure of the system in Section 4 through the production of a comparative study and some results from a real-world case study, and then the paper is brought to a close in Section 5.

## 2. Related Work

Researchers have made significant advancements in fault prediction for software systems on the industrial internet using deep learning algorithms. Several studies have explored the effectiveness of different approaches in enhancing fault prediction accuracy, precision, recall, and f-measure. Yang et al. (2019) [21] proposed a fault prediction model based on the combination of the locally linear embedding (LLE) algorithm and the long short-term memory (LSTM) algorithm. Their model was trained on datasets obtained from NASA’s MDP dataset and demonstrated superior performance compared to other existing algorithms. The authors emphasized the importance of effective dimension reduction and highlighted the benefits of leveraging deep learning techniques for fault prediction in software systems. A study by Barcelos and Cardoso (2021) [22] focused on bearing fault diagnosis using deep learning algorithms. They employed deep learning methods, including LSTM, to predict software faults. Their results showed that LSTM and other deep learning approaches outperformed existing models regarding accuracy and efficiency. Iqbal et al. (2019) [23] addressed the issue of fault detection and isolation in industrial processes using deep learning approaches. They analyzed previous studies, evaluated performance measures, and discussed commonly used datasets in software fault prediction. Their research highlighted the potential of data mining, machine learning, and deep learning techniques for improving software fault prediction. Another paper proposed a deep learning-based method for predicting industrial machinery’s remaining useful life (RUL) when only partial system health information is available [24]. The authors utilized a supervised attention mechanism to focus on informative data with significant degradation features while disregarding non-discriminative elements. Their approach aimed to provide effective and reliable machinery health assessment and prognostic methods in modern industries, considering the challenges posed by partial observations and practical restrictions.

Additionally, the authors in another study proposed a deep learning-based approach for predicting and diagnosing faults in superconducting systems [25]. The objective was to enhance the reliability and efficiency of fault prediction and diagnosis in superconducting technology. Their research focused on leveraging intelligent data-driven approaches to achieve promising prognostic results. These studies highlight the increasing interest in utilizing deep learning algorithms, such as LSTM, for fault prediction and diagnosis in software-intensive systems. The proposed models and approaches demonstrate improved performance compared to existing methods, emphasizing the potential of deep learning in addressing the challenges of fault prediction in various domains [26].

### 2.1. Fault Detection in Cold Forging Using Hybrid Models

A Convolutional Neural Network classifier was able to detect fault circumstances with a level of accuracy of 99.02% following the gathering of data from several faults that are frequently experienced, and it was also able to classify each fault with a level of accuracy of 92.66%. This was achieved after gathering data from several spots that are frequently experienced [27]. The findings indicate that deep learning may have the ability to identify flaws in cold forging. It is indisputable that systems are of the utmost significance, and their influence can be seen in virtually every area of contemporary life. In contrast, it is consistently expanding due to the transition of an increasing number of services to a digital format. Because of this, improving the procedures used to generate software and ensuring its quality is necessary to provide software that can be relied upon [28]. Consequently, a hybrid hidden Markov model in conjunction with an artificial intelligence model is used in the sensor dataset to carry out error detection. When it came to analysing live data and different gas mixtures, our technique performed far better than the conventional gas sensor array. The performance of our method was superior to that of other technologies that are now available, including monitoring potentially hazardous gases and identifying errors in sensor datasets. The hybrid HMM and ANN defect detection methods performed remarkably well on the datasets and had several false positives of 0.01% [29].

### 2.2. Fault Detection for Alarm Data Analysis

The LFOA method significantly improves the classification accuracy in a lightweight DCNN model designed for audio fault identification in cars. This approach reduces the number of neurons in the hidden layer of the DCNN and minimizes the number of input features extracted from the audio recordings. Implementing the LFOA algorithm makes it feasible to create a lightweight DCNN model suitable for implementation on edge processors like smartphones. Experimental results demonstrate that the suggested model enhances the accuracy of classifying the six faults to be identified, making it an effective research model for determining the health state of cars [30]. The novelty of this research resides in the fact that it proposes a novel AOC–ResNet50 network and its successful use in wind turbine defect detection. This was validated by a study that analyzed the detection of faults in wind turbine power converters in comparison to other competitive convolutional neural network models for deep learning [31]. The findings demonstrate that our neural network can predict numerous cells’ voltage despite varying degrees of degradation. In addition, it reduces the prediction error produced by the parametric model by a whopping 53%. Because of this enhancement, our network could forecast a fault 31 h before it actually occurred; this is a 64% increase in reaction time compared to the parametric model [32]. After that, an identity BiLSTM-CNN classifier is utilized to learn the organization and relevance between earlier alarm data. This step takes place after the preceding step. Following the completion of the training phase, the model is put to use for online fault detection. In conclude, the suggested model is used to analyze the well-known Tennessee Eastman process, and the results of using this model are shown [33].This is the case even if the sample has high inequality. This is exemplified by the fact that the method can still attain this degree of accuracy with some limitations. Compared with existing procedures considered to be state-of-the-art, the recommended strategy can improve diagnostic performance by somewhere in the neighborhood of 10% [34]. In addition to this, it produces high fault classification accuracy when applied to a complicated nonlinear rubbing fault signal. This finding leads one to believe that the proposed framework is especially appropriate for application in the massive factories that exist in the actual world [35].

## 3. Proposed Work

The gathering of data is the first step in the entire process of managing faults. This step entails the preprocessing of the data and the handling of features before training algorithms for defect detection and fault classification. After that, the detected faults are prioritized, and then either the operator’s staff or the automated system that handles the fault modification procedure will deal with them manually or automatically after that. This poll does not include all areas of inquiry due to the individualized nature of the steps or the level of maturity required for some of them. Gray boxes represent currently investigated study fields, whereas white boxes represent those yet to be explored. Figure 1 illustrates the architecture of the proposed model.

### 3.1. Data Acquisition Module

The data warehouse will be used for diagnosis and prognosis once the data gathering module has been completed, and all the data acquired will be stored there.

### 3.2. Data Preprocessing Module

However, suppose there is an excessive amount of unnecessary and redundant information during the process of knowledge discovery, such as noise or data that cannot be trusted. In that case, the training phase will be more difficult. As a result, performing some form of data preprocessing before moving on to the next phase is essential. In the business world, this obstacle is sometimes referred as BD. Cleansing the data, integrating the data, reducing the amount of data, and transforming the data are often the most important stages involved in the data preprocessing stage. The act of finding and fixing inaccurate or corrupt records from a database is referred to as “data cleaning”. This process involves filling in missing values, smoothing noisy data, identifying and deleting outliers, and resolving discrepancies, among other things. The process of integrating data involves merging information held in a number of different data stores. Integration done with care can assist in decreasing and eliminate redundancies and inconsistencies in the data collection that is produced as a result. Data reduction results in a representation of the data set that is substantially lower in volume but yet has the potential to generate the same or almost the same analytical conclusions as the original data set. There are a variety of approaches to dimensionality reduction. Among these methods is the straightforward approach of applying feature extraction methods to the data set. These methods extract features that are characteristics of an impending failure or fault from preprocessed signals. Among these methods is the straightforward approach of applying feature extraction methods. The features can often be derived from one of three domains: the time domain, the frequency domain, or the time-frequency domain. In the data transformation process, the data are either transformed or consolidated into forms suited for DM. This allows the DM process to be more effective, and the patterns obtained potentially allow for better comprehension. During the process of data capture, huge amounts of data are produced due to the development of storage medium and the ability to compute. The raw data can be successfully cleaned up by data preprocessing, the data dimensions can be reduced, and the data can then be stored back in the warehouse for knowledge discovery. As a result, large amounts of data can be transformed into features or statistical values before being used as input variables in the DM process.

### 3.3. Diagnosis/Prognosis Module

The method developed by Chen was utilized in order for us to discover the correlations that existed between the various sensor data sequences. For instance, the sensor data and is linked to a cluster of faults that are represented by the formula DT_a_ = d_i1_, d_i2_, …, d_im,_ while the sensor data b is related to a cluster of faults that is represented by the writing DT_b_ = dj_1_, dj_2_,…, dj_n_. Both of these notations refer to the same set of problems. The parallels between DT_a_ and DT_b_ are then estimated as the likeness of l_a_ and l_b_. This relationship is denoted by the symbol LS(l_a_,l_b_).

(1)
$$LS\left(l_{a},l_{b}\right)=\frac{\sum_{i=1}^{m}\;\;\underset{1\leq j\leq n}{max}\;\left(DS\left(d_{ai},d_{bj}\right)\right)+\sum_{j=1}^{n}\;\;\underset{1\leq i\leq m}{max}\;\left(DS\left(d_{bj},d_{ai}\right)\right)}{m+n}$$

where DS(da_i_, db_j_) represents the semantic relationship of fault with both d a_i_ and d b_j_, each of which individually relates to DT_a_ and DT_b_. The maximum number of errors remedied by the DT_a_ and DT_b_ is denoted by the “mandn” digits, which are placed in parentheses. Matrix LR (n lnl) are used to depict the parallels between various sensor data, and the value of Lij is used to indicate the degree of similarity between two distinct sensor data sets (li and lj).

In the next section, they will discuss the diagnostic approach for identifying the concealed models of sensors and data-related links and predicting the data connected with faults. In the first stage of the building phase, a feature set was crafted by first including the parallels, links, and relationships between sensor data and problems. This was done so that the set could be used. As a consequence of this, the building of a capability that is based on dual neural network convolutional and focus processes has taken place. The incorrect side of the system is in charge for learning the generic interpretations of an inertial sensors link, while the correct side of the system is responsible for learning the more significant linking links between wearable sensors and faults. After that, these two representations are integrated with the help of an additional convolution and closely linked layer. The connection rating of this representation was generated from the probability that a sensing data set is associated with an error. We will use the example of the sensor l1 and the defect d2 to demonstrate our method to dual CNN or the diagnostics associations.

### 3.4. Fault Prioritization Module

The fault prioritization module is crucial in fault handling and management systems across various industries, including manufacturing and medical imaging. Its primary objective is to quickly identify and prioritize faults or malfunctions that require immediate attention and resolution. The module can analyze and interpret large volumes of sensor data collected from production lines or imaging systems by utilizing data-driven methods, such as machine learning algorithms. This data analysis aids in reducing downtime, minimizing manufacturing costs, and enhancing overall productivity and quality. The fault prioritization module plays a vital role in optimizing operational efficiency by efficiently identifying and addressing critical faults, thereby improving the overall performance and reliability of the systems.

### 3.5. Experts Interaction Module

Fault prioritization is crucial in efficient fault management and quick amendment of faults in production lines. It is a process that involves the classification and prioritization of faults based on their nature, impact, and location in the manufacturing process. The goal is to accelerate the repair actions by personnel, reduce machine downtime, and minimize manufacturing costs. While data-driven methods and machine learning techniques can support fault management by utilizing sensor data to monitor machine health status, the literature suggests fault prioritization lacks research on available learning methods. Therefore, expert opinions become essential in determining the priority of faults and optimizing the fault amendment process.

### 3.6. Query Processing Module

Query Processing Module fault prioritization involves determining the order in which faults or issues within the module should be addressed based on their significance and impact on the system’s overall functionality. The prioritization process aims to optimize resource allocation and ensure efficient resolution of faults. Various factors can be considered when prioritizing faults, including the severity of the issue, its potential impact on system performance, the frequency of occurrence, and the level of effort required for resolution. By systematically evaluating and categorizing faults based on these factors, the development team can allocate resources effectively and address critical issues first to minimize disruptions and enhance the system’s stability and reliability.

### 3.7. Construction of Feature Matrix

Three biological axioms are combined to generate the feature map that compares the sensor datal 1 and the fault d2. To begin, a higher likelihood of a connection between l_1_ and d_2_ exists if both l_1_ and d_2_ exhibit homology and connection links with more prevalent sensor data. For example, if l_1_ and l_2_ perform comparable tasks and d2 was shown to be connected to l_2,_ then it is likely that l_1_ is also related to d_2_. Let’s say that x_1_ is the first row of L, assuming that it includes all of the connections that can be found across l_1_ and other RNAs in ln_2_. In the second column of D, which is labeled x_2_, we keep track of the relationships among d_2_ and each of the sensor data. x_1_ and x_2_ are combined to create a matrix with the dimensions of 2n. Second, the likelihood of l1 being connected to d_2_ increases if both disorders l_1_ and d_2_ share resemblance and relationship linkages of more prevalent issues. The relationships among l_1_ and each fault are listed in A’s first row, denoted by the letter x_3._ x_4_ is the second row of D, and it details various parallels that may be drawn among d_2_ and these disorders. In addition, x_3_ and x_4_ are merged, and the resulting matrix has the dimensions of 2nd. Third, a linkage among l_1_ and d_2_ is possible when they both interact and associate links with the shared sensor data. The contacts that occur amongst l_1_ and the different sensor data are recorded in the first row of Y, x_5_, and the second line of B, x_6,_ captures the relationships involving d_2_ and these sensor data. The product of integrating x_5_ and x_6 y_ields a vector with the dimensions 2n*m. When all of these cubes are joined together, the result is a features vector of sensor datal*1 and fault d2 with a height of 
$2\times \left(n_{l}+n_{d}+n_{m}\right)$
.

### 3.8. Convolutional Module on the Left

In order to train universal shallow representations for l_1_ & d_2_, the characteristic matrix consisting of l_1_ and d_2_, P, is fed into to the multilayer component located just on left. Because it is easier to understand with instances, I will use the first training sample, the first dense layer, so explain how well the combination or the pooled processes work. To acquire the maximum knowledge of P, we first generate a new matrix that we will call P’ by padding it with 0.

### 3.9. Convolutional Layer

The value 
$n_{f}$
 denotes the height of a filters within the initial convolutional, while its wide is denoted by 
$n_{w}$
. When applying the filters 
$W_{{\mathrm{conv}}_{1}}$
, 
ℝnw×nf
 to the matrices 
$P^{\prime }$
, one can obtain the cnn model by 
$Z_{{\mathrm{conv}}_{1}}$
, 
ℝconv1n

$\left(1,\;4\;-n_{w}\;+\;1\right),\left(1,{\;n}_{t}\;+\;2\;-{\;n}_{f}\;+\;1\right),$
 if the number of pixels is 
$n_{{\mathrm{conv}}_{1}}$
. 
$P_{i,j}^{\prime }$
 is the aspect that is located in the 
$i^{\mathrm{th}}$
 row and the 
$j^{\mathrm{th}}$
 article of 
$P^{\prime }$
 and 
$P_{k,i,j}$
 is an area that is included inside the filtration that is reached whenever the 
$k^{\mathrm{th}}$
 filters slipped to the location 
$P_{k,i,j}^{\prime }$
. The following are the official definition of the expressions 
$P_{k,i,j}$
 and 
$Z_{{\mathrm{conv}}_{1,k}}\left(i,j\right)$
:
(2)
$$P_{k,i,j}^{\prime }=P^{\prime }\left(i:i+n_{w},j:j+n_{f}\right),NP_{k,i,j}^{\prime }\in R^{n_{w}\times n_{f}}$$


(3)
$$Z_{{conv}_{1,k}}\left(i,j\right)=f\left(W_{{\mathrm{conv}}_{1}}\left(k,:,:\right)*P_{k,i,j}^{\prime }+b_{{conv}_{1}}\left(k\right)\right)$$


(4)
$$i\in \left[1,4-n_{w}+1\right],j\in \left[1,n_{t}+2-n_{f}+1\right],k\in \left[1,n_{{conv}_{1}}\right]$$

where 
$b_{{\mathrm{conv}}_{1}}$
 is the bias vector, 
f
 is a relu function, and 
$n_{t}=n_{l}+n_{d}+n_{m}.Z_{{conv}_{1,k}}(i,j)$
 is the element at the 
i
 th row and jth column of the 
k
 th feature map 
$Z_{{conv}_{1,k}}$
. 

#### Pooling Layer

Figure 2 explores the dual-CNN, a convolutional neural network (CNN) architecture that utilizes two CNN models to perform specific tasks or enhance the performance of a given task. The concept of dual-CNN has been explored in various domains, such as image captioning and image denoising. In these studies, dual-CNN models were designed and trained to address specific challenges and improve the performance of the respective tasks.

Pooling is a technique commonly used in CNNs to reduce the spatial dimensions of feature maps while preserving important information. It helps to extract the most relevant features and reduce computational complexity. Pooling layers, such as max or average pooling, are applied in a CNN architecture after convolutional layers.

Flattening refers to converting multidimensional data, such as feature maps in a CNN, into a one-dimensional vector. This transformation enables the input to be fed into a fully connected layer, a dense neural network layer. Flattening is typically performed before the dense layers to enable the network to make predictions or perform classification tasks.

Dense layers, also known as fully connected layers, are a type of layer in a neural network where each neuron is connected to every neuron in the previous layer. These layers are crucial in learning complex patterns and making predictions based on the extracted features. They allow the network to capture high-level representations and perform classification or regression tasks.

Recursive refers to a process or algorithm that repeats itself, typically with the output of one iteration becoming the input for the next iteration. In neural networks, recursive algorithms can be used for tasks such as sequence generation or processing hierarchical structures. Recursive neural networks have been applied in various natural language processing tasks, such as sentiment analysis and parsing.

Mixing can refer to different techniques or operations applied to neural networks. In the context of CNNs, mixing can refer to operations like concatenation or element-wise addition that combine features or representations from different layers or branches of a network. Mixing operations can enhance the network’s ability to capture diverse information and improve performance.

In the given context, stress does not have a specific meaning related to neural networks or deep learning. However, stress is a common term used to describe mental or emotional strain experienced by individuals. In the context of human well-being, stress management techniques or interventions can be explored to alleviate stress and promote overall health and productivity.

In max pooling, a two-dimensional filter is applied to each channel of the feature map, sliding over it and summarizing the features within the region covered by the filter. The output dimensions of a pooling layer can be calculated as 
$Z_{\mathrm{convpool}1}$
,

(5)
$$Z_{\mathrm{convpool}1,k}\left(i,j\right)=\mathrm{Max}\left(Z_{conv1,k}\left(i:i+n_{g},j:j+n_{p}\right)\right)$$


(6)
$$i\in \left[1,5-n_{w}-n_{g}+1\right],\;j\in \left[1,n_{t}+3-n_{f}-n_{p}+1\right]$$


(7)
$$k\in \left[1,n_{\mathrm{conv}\;1}\right],$$

where 
$Z_{\mathrm{convpool}1,k}$
 is the 
k
 th feature map, and 
$Z_{\mathrm{convpool}1,k}(i,j)$
 is the element at its’ 
i
 th row and jth column.

### 3.10. Attention Module on the Right

In our concept, the focus modules are responsible for determining whether characteristics or connection links are relevant for the depiction of sensor data-l one and fault d2. The proposed technique at the classification stage and the one at the person level are thus both components of the module.

In most cases, unique characteristics within P provide distinct contributions to various sizes and types of sensor data and their associated faults. For example, in relation to a particular fault, the sensor data that were shown to be connected with fault are frequently more significant than those that have not been found to be related to fault. Every component x_ij_ of vector x_i_ is given an attentive value referred to as _ijF in the square matrix P = x_1_, x_2_, …, x _i_, …, x_6_. The attentive value _ijF is as follows:
(8)
$$s_{i}^{F}=H^{F}\tanh\left(W_{x}^{F}x_{i}+b^{F}\right)$$


(9)
$$\alpha_{ij}^{F}=\frac{\exp\left(s_{ij}^{F}\right)}{\sum_{k}\;\;\exp\left(s_{ik}^{F}\right)}$$

where in HF and W xF are the frequency vectors, and bF is an error signal for the whole function. si^F = [s_i1_^F, si_2_^F, …, s_ik_^F, …, s(in_i_)^F] is the vector that stores the attentiveness scores that signify the significance of various characteristics included in x_i_, where in n_i_ is the duration of x_i_ and s(in_i_)F is the value that x(in_i_) has been given. The normalized attentiveness value for variable x_ij_ is denoted by the symbol _ijF. Therefore, the hidden expression of the various characteristics may be represented by the symbols y_i_,

(10)
$$y_{i}=\alpha_{i}^{F}⊗x_{i}$$

where 
⊗
 is the element-wise product operator, and the symbol 
F
 represents the feature level.

### 3.11. Attention at the Relationship Level

There are a number of different link relationships which exist among sensor data and faults. These correlation partnerships also include commonalities between circRNAs, the same agreements that exist with both sensor data & faults, the resemblances among faults, this same interplay that exist with sensor data and sensor datas, and the affiliations between faults and sensor data. The portrayal of Sensor data-fault connections also is affected differently according to the various correlations that are considered. Thus, to build a full attention representation at a real level, you apply a learning algorithm to each vector y_i_ separately. The following factors contribute to the attentiveness ratings at the person level:
(11)
$$s_{i}^{R}=h^{R}\tanh\left(W_{y}^{R}y_{i}+b^{R}\right)$$


(12)
$$\beta_{i}^{R}=\frac{exp\left(s_{i}^{R}\right)}{\sum_{j\in 6}\;\;exp\left(s_{j}^{R}\right)}$$

where in W_y_R represents the set of weights, and b^R^ represents the biased vector. h^R^ is indeed a scale parameter, and s i^R^ is the scores of the ith connection y_i_. i^R^ is the attentive value that has been normalized for connection y_i_. The gained and expressed hidden depiction of connection is achieved by the affections at the component and relation levels.

(13)
$$g=\sum_{i}\;\beta_{i}^{R}y_{i}$$

where the letter R denotes the degree of the existing connection. Let G represent the grid following g has been filled with zeros to pad it. After putting G into a convenient and maximum pooling layer, the attentiveness visualizations Z “att” may then be generated.

### 3.12. Final Module

Let’s call the represented learnt from the left recurrent module Z glo and the information acquired from the middle convolution modules 
$Z_{\mathrm{att}}$
. 
$Z_{\mathrm{att}}$
 will be the information that was learned about focus. The combination of Z glo and 
$Z_{\mathrm{att}}$
 is symbolized by the symbol 
$Z_{\mathrm{con}}$
 which is formed by placing a first on top of the latter and placing the latter below it (Figure 2). In order to produce the final form 
$Z_{\mathrm{fin}}$
, the 
$Z_{\mathrm{con}}$
 level is subjected to an extra neural processing step. 
$z_{0}$
 is a vector created by flattening Z fin, and it is input into a convnet called W out as well as a softmax to generate the value p.

(14)
$$p=\mathrm{softmax}\left(W_{\mathrm{out}}z_{0}+b_{0}\right)$$

pi is an associated random variable of C subclasses, with C equal to two, so it includes both the likelihood that a condition and sensor data are found to have an associated link and the likelihood that they do not have a companion.

#### Loss of Association Diagnosis

In our concept, the bridge loss between the test dataset distribution of lncRNA-fault association and the diagnostic probability p is defined as L, where L is the bridge loss between the test dataset distribution and the probability of diagnosis.

(15)
$$L=-\sum i^{\mathrm{TX}}*\sum {\left(j=1\right)}^{{\mathrm{Cz}}_{j}\log p_{j}}$$

where the classifying label vector is denoted by zR2 and the set of training instances is denoted by T. If l_1_ is connected to d_2_, then the vector z has a dimension of 1 for its second dimension while it has a dimension of 0 for its first dimension. On the other hand, if l_1_ is not connected to d_2_, then the initial aspect of z is equal to 1, while the second dimension is equal to 0.

We refer to all of the parameters of the neural network as. The following is a definition of the objective function that plays a role in our learning process:min┬θ L(θ) = L+λ‖θ‖^2(16)
where is a parameter that represents a trade-off between the amount of the training sample and the regularisation term. We employ the Adam optimization technique to achieve maximum efficiency in the objective function.

### 3.13. Data Acquisition and Signal Processing and Analysis

Engineers can access fundamental components such as analog-to-digital converter boards (DAQ boards) and software like LabVIEW in data acquisition systems. These tools empower engineers to design and create their customized data acquisition systems. With LabVIEW’s capabilities, engineers can perform data processing, analysis, and real-time display on their computer’s monitor efficiently and effectively. LabVIEW, in conjunction with the NI-DAQmx driver, facilitates the development of triggered applications for data acquisition. The NI-DAQmx applications typically consist of four building blocks, and trigger conditions for these applications are specified in the configuration section through appropriate function calls or VIs in LabVIEW.

### 3.14. Fault Prioritization Module

To proceed to the following phase, which is prioritizing defects, it is necessary to first classify and discover problems in the data according to their different kinds. The perspectives of knowledgeable individuals are required for the phase of prioritizing. They need to take into account the many potential problems that may arise in the sensed information. They must also consider that machines do not operate alone but rather depend on one another. Therefore, in addition to the importance of individual machines, industry professionals need to consider the chain of failures and how machines impact other operations along the production line. The expert level of domain knowledge is also very crucial. Both the assembly lines and the CPSs that are used possess a high level of originality and provide a wide variety of possible configurations. It is difficult to perform fault prioritization in an industrial setting due to the scarcity of algorithms that do not rely on statistics. Fuzzy logic is an efficient strategy that may be utilized in many contexts to lessen the negative effects of criticized drawbacks. The approach has been tested and shown to be capable of operating in a variety of different industrial settings. However, this strategy relies on statistics and is time-consuming and labor-intensive, making it a costly choice. Due to the high expertise and familiarity required with the subject matter, the procedure cannot be automated. Some preliminary attempts have been made to build an automated fault prioritizing system. However, they do not pertain to Industry 4.0. In the realm of manufacturing, we see a significant opportunity to incorporate information from other fields and fields of study. This is of the utmost importance since digitization typically increases the complexity of manufacturing facilities, and the only thing that may minimize supply interruptions is effectively prioritizing.

### 3.15. Query Processing and Interaction with Experts/Consumers

Natural language processing (NLP) is a methodical computer technique that enables the acquisition of information about how people use and understand language. It has advanced significantly with the help of artificial intelligence, contributing to Industry 4.0, which represents a new level of innovation, technological advancement, and economic prosperity. The concept of Industry 4.0, first introduced by the German government in 2011, refers to an economy characterized by advanced industrialization and has gained global recognition. The integration of NLP and artificial intelligence, along with the Internet of Things (IoT), has brought humans closer to realizing Industry 4.0, enabling global participation and benefits. NLP and AI have facilitated improved communication, personalized goods and services, and influenced customer choices by understanding their requirements. The study uses pragmatic analysis to analyze data context, including attitude, intention, and fault representation. Industries utilize global optimization based on globalization to achieve significant categorization. Industry 4.0 is tightly integrated with supply chain management, emphasizing communication as a key focus for modern enterprises. NLP applications are crucial in Industry 4.0, supporting various communication channels and enhancing business operations and services. The combination of AI and NLP allows businesses to understand their customers better, meet their needs, and determine customer requirements based on company priorities.

## 4. Experimental Results & Discussion

Accuracy, Precision, Recall & F1-score are the four measures offered for the assessment in the study. Correctness was the most important parameter. First, we will refer to factual favorable, untrue optimistic, true bad, and systematic error in both TP, FP, TN, and FN. An integral gain is the number of times in which a prediction was accurate when it was necessary, a falsified pleasant is the number of times in which a prediction was erroneous when it was necessary, true negative seems to be the number of occasions in which a prediction was correct when it was not needed, and a false negative is the number of cases in which a prediction was inaccurate when After that, the following is how we come up with all those four factors: efficiency, clarity, recollection, and f1 score.

(i)Accuracy

The term “precision” refers to the degree to which retrieved data matches the originals, expressed in percentage (%), from a data base (such as a cloud server architecture, for example).

(17)
$$Accuracy=\frac{TP+TN}{TP+FP+TN+FN}$$


(ii)Precision

The ratio of accurate diagnoses to total diagnoses is one way to measure the amount of detail in a test. It substantiates the capacity to obtain the highest-ranked papers that contain the most pertinent information.

(18)
$$Precision=\frac{TP}{TP+FP}$$


(iii)Recall

The memory percentage may be calculated as a ratio of positive instances to TP + TN. It provides more evidence that the search can identify all of the objects on the database that are correct.

(19)
$$Recall=\frac{TP}{TP+FN}$$


(iv)F1-Measure

It is the weighted harmonic mean of the precision and recall, and it represents the overall performance.

(20)
$$F1-Measure=\frac{2\times Precision\times Recall}{Precision+Recall}$$


In addition to the performance metrics discussed before, a Operating Characteristic (ROC) curve is another tool that can be used to assess the benefits and drawbacks of the diagnosis system. The Curve illustrates the compromise that must be made in between true positives (TPR) and the false positive rate (FPR). The terms “true positive” and “true alarm rate” are described as follows:
(21)
$$TPR=\frac{TP}{TP+FN},\;TFR=\frac{FP}{FP+TN}$$


Whenever the ROC curve is closer to the top left corner of the graph, the model is considered higher quality. In the health system we are proposing, each of these four metrics receives greater focus. When the memory and accuracy rates are both high, there is less possibility that a person who will be at danger for a sickness will be anticipated to have no major dangers in healthcare. This chance decreases as the accuracy rate and the result increase considerably. There is a high precision, recall, and F1-measure level, with respective values of 98%, 97%, 98.5%, and 98.6%. Table 1 presents the quantitative examples comparing the methods to those already in use.

In Table 2, increasing the number of epochs allows the model to learn more complex patterns in the data, potentially improving its performance. However, it’s important to monitor the model’s performance on a validation set during training and stop training when the validation performance starts to decay to avoid overfitting.

In Table 3, the batch size is a key parameter in machine learning, specifically in deep learning an d machine learning. It determines the number of training examples processed before updating the network weights and affects the trade-off between computational efficiency, stability, and convergence speed of the learning algorithm.

Table 4, dropout is a regularization technique used in machine learning, particularly in deep neural networks, to address the problem of overfitting. It involves randomly dropping out a subset of nodes during training, forcing the web to learn more robust representations and improving its ability to generalize to new, unseen data.

Table 5, in this architecture, the hidden layer is the layer between the input and output layers, containing a set of hidden units or neurons. Each neuron in the hidden layer receives inputs from the input layer, performs a weighted sum of these inputs, applies an activation function to the sum, and then passes the result to the next layer.

Table 6, shows multiple hidden layers in a neural network, and deep learning models play a crucial role in learning complex tasks and capturing intricate relationships within the data. By stacking hidden layers between the input and output layers, a neural network can process the data through multiple stages of computation, allowing for the extraction of higher-level features and patterns.

Table 7, ReLU and sigmoid activation functions have distinct characteristics that make them suitable for different scenarios. ReLU is computationally efficient, helps alleviate the vanishing gradient problem, and promotes sparsity, making it popular for hidden layers. On the other hand, sigmoid provides a probabilistic interpretation, which is useful for binary classification tasks and produces smooth outputs. It is important to consider these factors when selecting an activation function based on the specific requirements of deep learning models.

In Table 8, these statistical measures are widely employed in deep learning to assess model performance, understand data distributions, and make informed decisions during the training and evaluation process.

In Table 9, it is important to consider both training and testing accuracies when comparing different models. High training accuracy indicates that the model has learned the patterns in the training data, while high testing accuracy suggests that the model can generalize well to unseen data. A significant drop in testing accuracy compared to training accuracy could indicate overfitting, which needs to be addressed through regularization techniques or model adjustments.

Table 10, the Silhouette Coefficient, is a valuable metric for assessing the quality of clustering algorithms. It measures the coherence and separation of clusters, providing insights into the effectiveness of the clustering process. Its calculation involves determining the average distances within and between clusters. By applying the Silhouette Coefficient, we can evaluate the tightness and distinctiveness of the clusters and make informed decisions in unsupervised learning tasks.

Table 11, Accuracy is a commonly used measure for evaluating machine learning models, particularly in classification tasks. It represents the fraction of correct predictions made by the model out of the total number of predictions. However, accuracy alone may not provide a complete picture of model performance, especially when dealing with class-imbalanced datasets. Class-imbalanced datasets have a significant disparity between positive and negative labels.

### 4.1. Impact on Accuracy

Quantifying it as the biggest equivalent gap between data and actuality is possible. It ought to be set at medium since it works out well there. In contrast hand, precision describes how closely the test result corresponds to the true worth. Figure 3 displays the results of the measurements used to determine the precision. The effect that varying the volume of data stored on the public cloud has on the success rate is seen in Figure 3. These findings indicate that the reliability of retrieving data and classifying it on the cloud platform improves in proportion to the magnitude of the amount of data stored on the public cloud. This is because permanent nodes are more likely, resulting in correct judgments in the suggested technique. Old attempts like CNN and I-CNN and the proposed model had a tendency to boost the number of requests that needed to be done while simultaneously decreasing the quantity of data that could be retrieved. This significant departure from the overall average may lead to bad findings comparable to those obtained before as a consequence of the diagnosis and prognosis. Consequently, it may be challenging to recover precise findings about the rectification and analysis of data.

### 4.2. Impact on Precision

The diagnostic of the customer may be validated with the assistance of expertise. It can be defined as the proportion of correct diagnoses produced relative to the grand total of diagnoses made. Figure 3 depicts the graphical representation of the accuracy value. Determining the appropriate sample size is a crucial aspect of research design. It requires considering the sample’s representativeness, statistical power, ethical considerations, and specific research objectives. While sample size formulas and guidelines can provide initial estimates, it’s essential to tailor the sample size to the specific research context and population of interest Figure 4.

### 4.3. Impact on Recall

During data acquisition, the Forget meter analyzes the nature of the matter and discovers as much supporting paperwork as possible. Figure 5 compares the results of the. It is a performance metric that considers both accuracy and memory and is known as the harmonized mean. The harmonized mean of two integers often gets closer to the less significant value of the two. For the harmonized mean to just be high, it is necessary for both the accuracy and the memory to be high.

### 4.4. Impact on F1-Measure

In Figure 6, the concepts of recall and accuracy are opposed. It is typical for recall to decrease as accuracy increases. The value of a Sound and effective is at its highest at 1 and its lowest at 0. Figure 6 is a comparison of the many machine learning techniques that have been used in earlier methods as well as those that have been developed by modifying the size of the sample.

The outcome of the suggested technique achieves a greater quality than both CNN, I CNN, SVM and proposed model. The methods get satisfactory results, as shown in Figure 5, but their results are inferior to those achieved by the suggested approach. These findings suggest that the effectiveness of the suggested method will be adequate for retrieving health data and identifying health issues.

### 4.5. Results of ROC Curve

Because it summarizes the result of diagnostics, it’s more crucial taking. To construct a curve, the diagnostics criteria is permitted to fluctuate. The Curve considers only instances with a proper result and the quality of the result. Figure 6, depicts the outcome of the evaluation for the Curve. It contains a chart that presents the Residual plot that describes the performance of the forecasting model when applied to cloud server architecture. It represents the true positive rate and the false positive rate for the risk factors diagnosis and the correct data collection.

Our findings of my experiments have demonstrated that one may use a model again to recover associated health data, and another model can be used to diagnose faults using an engine that uses deep supervised learning. This model has produced adequate results regarding reliability, clarity, memory, and f1-score. These results were achieved by data preprocessing, diagnosis and prognosis, fault prioritization, expert interaction and query processing, and retrieval of queries.

### 4.6. Complexity Analysis

In this part, we will discuss the calculation of our presented model. Specifically, we will discuss the runtime, the parameter settings, and the restrictions in secure. The following is indeed a rundown of the time and place complexities involved in data recovery and strategic planning:

#### 4.6.1. Time Complexity

In the first step of this process, we determine the amount of time required to retrieve data. The process of finding the data involves a consistent amount of time. The request is performed during the user’s use, and then contrast is done with the unit time. The duration cost for the retrieving data is broken down as follows:
(22)
Time_Complexity=O(n)+O(q)


Again for administration and storage of large amounts of data using cloud services, certain significant processes take longer than others; nonetheless, they are pushing new techniques that deliver quicker work quality. Pre-processing stage, the collection of features, and diagnostics are all examples of this kind of procedure. The well-before method involves 
O(n
) room; this same lossless encoding process will take O(c) space, and the prognosis use someone 
O(p)
 room. This procedure should be implemented for each logbook diagnosis in a unit of time, and it takes time. Additionally, the whole procedure needs to perform on algorithm type, which is why it takes time.

(23)
$$Time\_Complexity=\alpha \left[O(n)+O(c)+O(p\times \gamma )\right]$$


#### 4.6.2. Space Complexity

In the procedure of retrieving, two aspects are taken into consideration for a patient’s specific query: the amount of data that is being requested and the querying procedures. The basic kind of space that is necessary for the other activities. The complexities of the space required for data retrieving may be expressed as follows:
(24)
Space_Complexity=O(n)+O(q)


In managing data, four components are running: preparation, quantization, and diagnostic. There is an O(n) need for room during or before the step, an O(c) need for space during the image compression step, and an O(n) need for storage during the diagnostic step (p). Units much area is required for various other operations, including data entry, verification, file division and keeping, computing, connection, and modeling for data exchange across health data. As a result, the following is an example of the relative spatial involved in data mangers:
(25)
Space_Complexity=O(n)+O(c)+O(p)


## 5. Conclusions

Artificial intelligence is being increasingly relied upon in industrial maintenance plans to anticipate the state of assets and prescribe appropriate maintenance measures. The associated software for maintenance and the human maintenance actors can combine to produce a hybrid-augmented intelligence system. In this type of system, each side of the partnership can learn from and improve the other side’s intelligence. This system requires human-machine interfaces that have been enhanced to assist users in expressing their expertise and retrieving information from software that is difficult to use. In addition, researchers should build and manage publicly useable training datasets to expand language models if the models do not have sufficient accuracy. As a result, this paper aims to suggest an innovative strategy for maintenance specialists and operators to connect with a predictive maintenance system through a digital intelligent assistant. This assistant is an example of artificial intelligence (AI), and it is designed to aid users in interacting with a system using normal language while also collecting feedback from users about the effectiveness of maintenance interventions. In light of this, a hybrid Convolutional Neural Network model is developed to diagnose the data obtained from the sensors. A strategy based on fuzzy logic is suggested for fault prioritizing. This approach prioritizes the defects according to the damages or costs they cause. The effectiveness of the presented strategy is also dependent on the persistent enhancement of understanding of natural language via updating data according to languages and also through the processing of queries. Conversation-driven development will be necessary for such a procedure. This is the type of development in which actual interactions with the assistance generate precise training data for language and dialogue models. In the future, recommendation strategy i.e., personalized approach, is presented for industrial applications. The novel defect diagnosis system for plaster production employs hybrid convolutional neural networks to support quality control employees in identifying the sources of deviations and implementing necessary corrective actions. The system integrates X-bar and R charts to detect multiple assignable causes simultaneously and recognizes non-random patterns to estimate parameters, change points, and factors responsible for abnormal patterns. It is important to note that while Adam generally performs well, it may not always be the best optimizer for every task.

In some instances, researchers have observed that switching to SGD can lead to better generalization performance. The prospects of using the Adam optimizer, Ridge Regression, and Feature Mapping include efficient optimization, adaptive learning rate, prevention of overfitting and multicollinearity, handling non-linear relationships, increased model capacity, and improved performance. Experimenting and choosing the techniques that best suit the specific problem is crucial. The practical uses of the Adam optimizer adapt the learning rate for each weight in a neural network by estimating the first and second moments of the gradients. Ridge Regression is a regularization technique used in linear regression to mitigate the problem of multicollinearity and overfitting. Feature mapping allows capturing non-linear relationships between the input features and the target variable by introducing non-linear transformations. Feature mapping can enhance the predictive performance of machine learning models by providing a richer representation of the data.

## Figures and Tables

**Figure 1 sensors-23-07011-f001:**
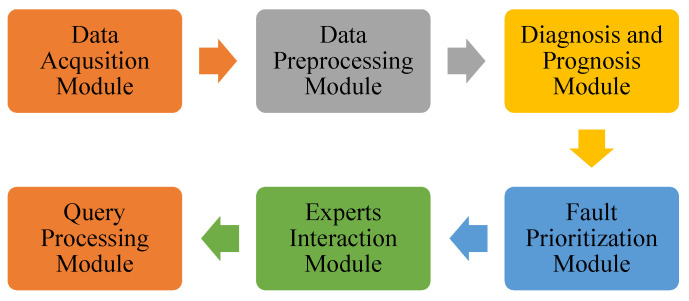
Architecture for Proposed Model.

**Figure 2 sensors-23-07011-f002:**
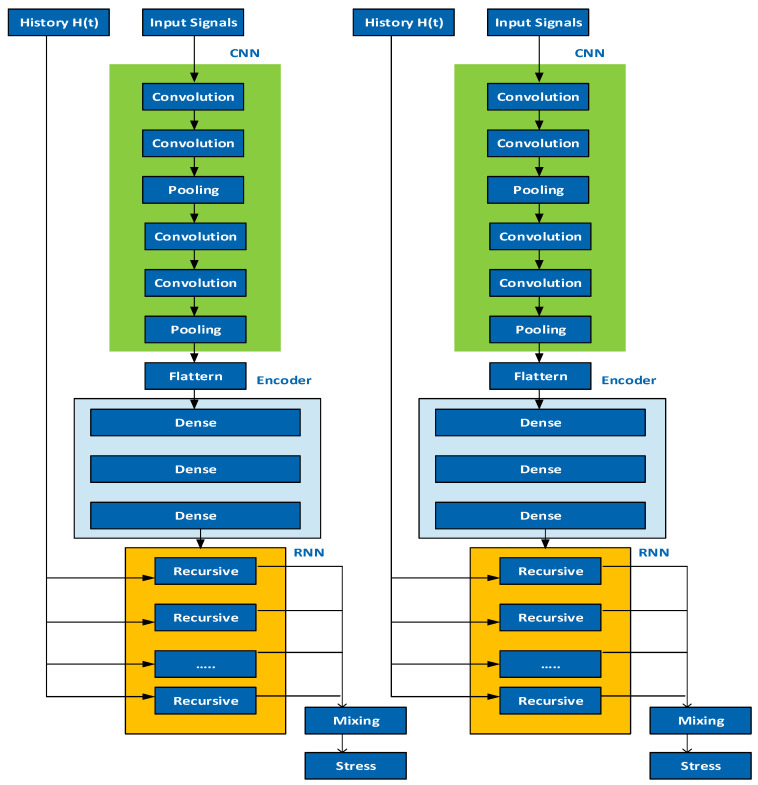
Dual CNN for Fault Diagnosis.

**Figure 3 sensors-23-07011-f003:**
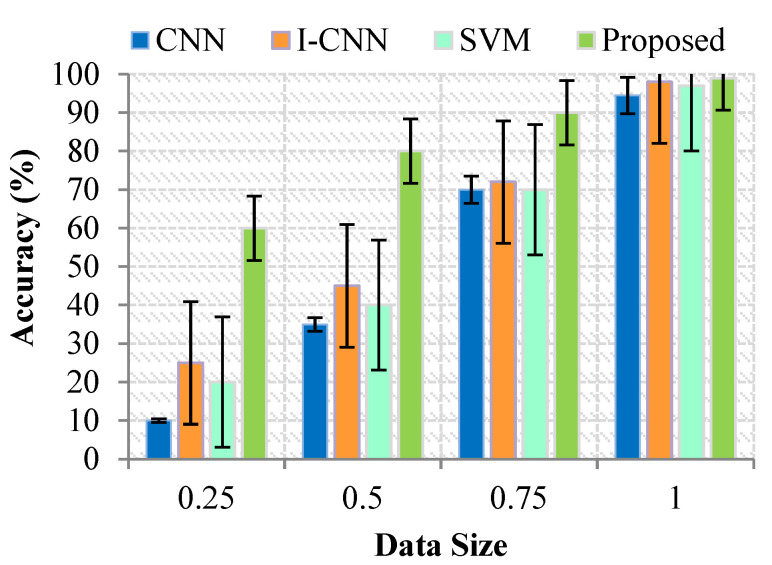
Results for Accuracy (%) with Data Size.

**Figure 4 sensors-23-07011-f004:**
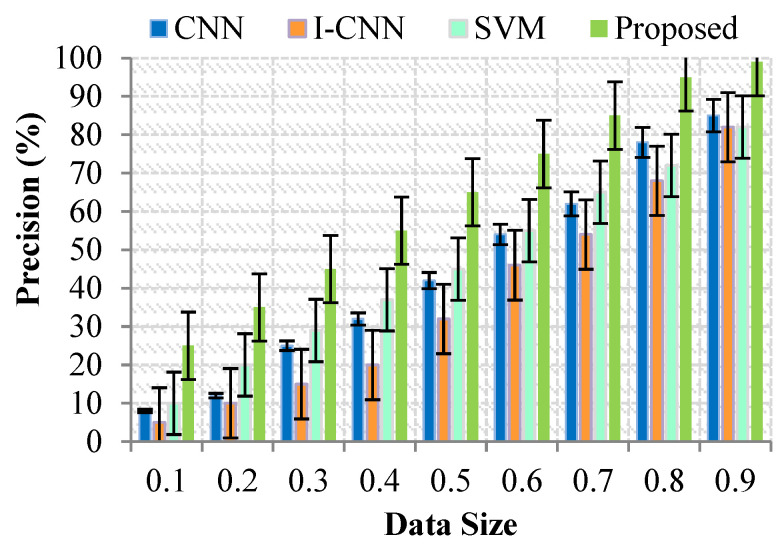
Results for Precision (%) with Data Size.

**Figure 5 sensors-23-07011-f005:**
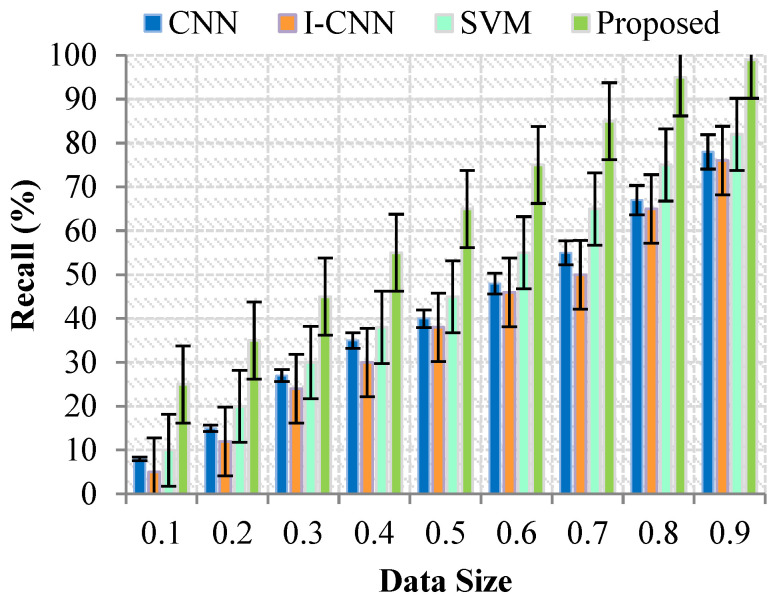
Results for Recall (%) with Data Size.

**Figure 6 sensors-23-07011-f006:**
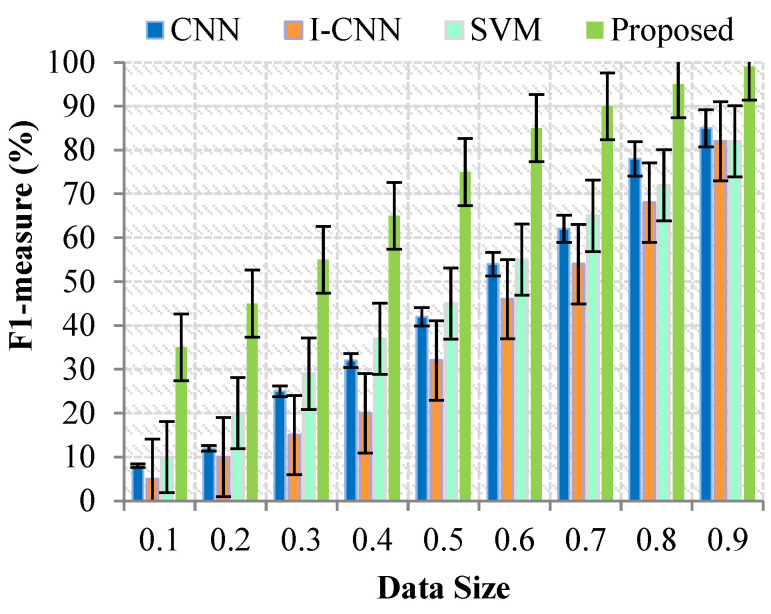
Results for F1-measure (%) with Data Size.

**Table 1 sensors-23-07011-t001:** Proposed and existing methods comparison.

Methods	Avg. Accuracy	Avg. Precision	Avg. Recall	Avg. F1-Measure
CNN	94.5%	91%	91%	91%
I-CNN	97%	92%	92.5%	94.5%
SVM	91%	92.39%	88.54%	90.43%
Proposed	98%	97%	98.5%	98.6%

**Table 2 sensors-23-07011-t002:** Proposed versus existing methods epochs.

Epochs	CNN				I-CNN			
	Accuracy %	Precision %	Recall %	F1-Score %	Accuracy %	Precision %	Recall %	F1-Score %
100	91.1	92.08	92.12	92.04	92.94	92.57	92.17	92.12
200	92.26	92.21	92.91	93.02	92.1	92.67	92.15	92.57
500	93.32	92.15	92.32	92.15	93.19	92.15	92.01	92.92
1000	93.61	93.34	93.42	93.38	93.78	93.16	92.97	93.04
1200	93.58	93.33	93.32	93.41	93.77	93.08	93.32	93.13
**Epochs**	**SVM**				**Proposed**			
	**Accuracy %**	**Precision %**	**Recall %**	**F1-Score %**	**Accuracy %**	**Precision %**	**Recall %**	**F1-Score %**
100	81.83	81.56	81.18	81.18	92.95	92.59	92.18	92.14
200	81.98	81.64	81.29	81.17	92.13	92.68	92.16	92.59
500	82.17	82.17	82.15	82.13	93.21	92.17	92.06	92.95
1000	82.22	82.21	82.18	82.18	93.81	93.18	92.98	93.07
1200	82.81	82.19	82.17	82.16	93.79	93.12	93.41	93.05

**Table 3 sensors-23-07011-t003:** Proposed versus other methods batch size.

Batch Size	CNN				I-CNN			
	Accuracy %	Precision %	Recall %	F1-Score %	Accuracy %	Precision %	Recall %	F1-Score %
200	91.1	92.02	92.12	92.03	92.92	92.55	92.17	92.12
400	92.18	92.13	92.91	93	92.99	92.64	92.15	92.56
500	93.55	93.27	93.24	93.46	93.77	93.17	93.33	93.17
700	93.16	93.15	93.04	93.36	93.19	92.79	92.97	92.96
**Samples**	**SVM**				**Proposed**			
	**Accuracy %**	**Precision %**	**Recall %**	**F1-Score %**	**Accuracy %**	**Precision %**	**Recall %**	**F1-Score %**
200	81.83	81.57	81.17	81.19	92.93	92.57	92.19	92.15
400	81.97	81.63	81.27	81.18	92.99	92.68	92.17	92.59
500	82.75	82.19	82.13	82.25	93.79	93.19	93.35	93.2
700	82.23	82.15	82.1	82.14	93.23	92.81	92.99	92.98

**Table 4 sensors-23-07011-t004:** Proposed versus other methods dropouts.

Dropouts %	CNN				I-CNN			
	Accuracy %	Precision %	Recall %	F1-Score %	Accuracy %	Precision %	Recall %	F1-Score %
25	91.09	92.03	92.09	92.04	92.93	92.56	92.17	92.13
50	92.18	92.14	92.91	93.01	92.98	92.65	92.15	92.57
75	93.56	93.27	93.32	93.47	93.76	93.38	93.32	93.81
100	93.07	93.26	93.24	93.18	93.18	93.17	93.19	93.16
**Dropouts %**	**SVM**				**Proposed**			
	**Accuracy %**	**Precision %**	**Recall %**	**F1-Score %**	**Accuracy %**	**Precision %**	**Recall %**	**F1-Score %**
25	81.83	81.56	81.16	81.18	92.95	92.58	92.19	92.15
50	81.98	81.63	81.27	81.17	92.99	92.67	92.17	92.59
75	82.74	82.21	82.15	82.26	93.79	93.39	93.35	93.83
100	82.16	82.17	82.11	82.09	93.21	93.19	93.22	93.19

**Table 5 sensors-23-07011-t005:** Comparison of Single hidden layer with samples.

Single Hidden Layer with Samples in %	CNN				I-CNN			
	Accuracy %	Precision %	Recall %	F1-Score %	Accuracy %	Precision %	Recall %	F1-Score %
25	92.79	92.15	92.15	92.27	93.04	93.03	92.99	93.03
50	93.15	92.79	92.81	92.48	93.27	93.17	93.13	93.14
75	**93.69**	**93.18**	**93.19**	**93.18**	93.45	93.15	93.25	93.26
100	93.51	93.02	92.99	92.99	**93.77**	**93.37**	**93.47**	**93.36**
**Single Hidden Layer with Samples in %**	**SVM**				**Proposed**			
	**Accuracy %**	**Precision %**	**Recall %**	**F1-Score %**	**Accuracy %**	**Precision %**	**Recall %**	**F1-Score %**
25	81.24	81.13	81.69	81.78	93.05	93.04	92.98	93.05
50	81.77	82.15	61.47	81.88	93.29	93.18	93.15	93.16
75	82.47	82.17	82.19	82.21	93.47	93.17	93.27	93.28
100	82.42	82.11	82.14	82.19	93.79	93.39	93.49	93.38

**Table 6 sensors-23-07011-t006:** Comparison of Multiple hidden layer with samples.

Multiple Hidden Layer with Samples in %	CNN				I-CNN			
	Accuracy %	Precision %	Recall %	F1-Score %	Accuracy %	Precision %	Recall %	F1-Score %
25	92.79	92.15	93.14	92.27	92.04	93.02	92.03	93.03
50	93.16	92.28	92.19	92.43	91.27	92.17	92.18	92.18
75	**93.67**	**93.35**	**93.32**	**93.14**	93.49	93.39	93.27	93.28
100	93.24	93.31	93.29	93.09	**93.53**	**93.46**	**93.32**	**93.31**
**Multiple Hidden Layer with Samples in %**	**SVM**				**Proposed**			
	**Accuracy %**	**Precision %**	**Recall %**	**F1-Score %**	**Accuracy %**	**Precision %**	**Recall %**	**F1-Score %**
25	81.25	81.13	81.69	81.79	97.06	96.04	97.05	97.06
50	81.27	82.26	81.47	81.91	96.29	95.19	97.19	96.19
75	82.59	82.37	82.18	82.19	97.52	96.41	98.29	97.29
100	82.43	82.34	82.14	82.17	98%	97%	98.50%	98.60%

**Table 7 sensors-23-07011-t007:** Comparison of activation functions.

Activation Function	CNN %	I-CNN %	SVM %	Proposed %
Sigmoid	94.5	97	91	98
Relu	92.23	94.56	90.23	92.43

**Table 8 sensors-23-07011-t008:** Statistical measures.

Models	Computational Time (s)	MSE	RMSE	R2
CNN	28.063	0.691	0.054	0.998
I-CNN	23.589	0.439	0.121	0.934
SVM	13.767	0.231	0.058	0.969
Proposed	29.921	0.195	0.062	0.897

**Table 9 sensors-23-07011-t009:** Training versus testing accuracy.

Models	CNN	I-CNN	SVM	Proposed
Accuracy for training data	0.932	0.985	0.869	0.999
Accuracy for testing data	0.944	0.97	0.883	0.996

**Table 10 sensors-23-07011-t010:** Interpret and validate the consistency within clusters of data.

Models	Silhouette Coefficient
CNN	0.937011
I-CNN	0.954649
SVM	0.968802
Proposed	0.966503

**Table 11 sensors-23-07011-t011:** Comparison of various existing methodologies.

Models		Accuracy%
K-Nearest Neighbor	(KNN)	74.09
Naïve Bayes	(NB)	72.91
Decision Tree	(DT)	72.98
Artificial Neural Networks	(ANN)	70.11
Multi-Layer Perceptron	(MLP)	73.13
Convolutional Neural Network	(CNN)	82.36
Radial Basis Neural Networks	(R-BNN)	78.31
Long Short Term Memory	(LSTM)	83.67
Bidirectional LSTM	(BILSTM)	84.89
AdaBoost	(AD)	71.56
BootStrapping	(BS)	71.34

## Data Availability

Implementation code: https://github.com/NARAYNAN888/industry_prediction/commit/ba2e23b744be17c6fc9d220008d1e0ac06f6acd8; Implementation code document (accessed on 25 April 2023). https://github.com/NARAYNAN888/industry_prediction/blob/main/Cnn%20model%20for%20industry%20prediction.docx; Dataset from kaggle (accessed on 20 June 2023). https://www.kaggle.com/code/koheimuramatsu/model-explainability-in-industrial-image-detection/input (accessed on 20 June 2023).

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
