# Peer review of "Enhanced Dual Convolutional Neural Network Model Using Explainable Artificial Intelligence of Fault Prioritization for Industrial 4.0"

_sensors, 2023, doi:10.3390/s23157011_

Round 1

Reviewer 1 Report

What kind of software was employed for data acquisition and signal processing and analysis? Did you use in-house code (self-developed) or any other commercial tools for data mining?

English expressions and spelling typos should be reviewed

Reviewer 2 Report

The paper contains an interesting piece of work. The works suggested by the author is a very good contribution for an innovative defect diagnosis system for plaster production that utilizes hybrid convolutional neural networks to assist quality control employees in identifying the sources of deviations and taking necessary corrective actions. But, based on the previous networks, the paper's subject is relatively simple and old in this research area although the analysis and evaluation of the works seems to be sufficiently done. An additional limitation is that the writing does not appear to provide a novel contribution to the state of the art. 

What is the your novel main idea?

In particular, this lack of logical configuration is confusing in Section 1. As far as I can see, some sub-sections of this section describe the relevant works. These must be done in revised version separately.

Section 2 could be better presented as a sub-section, and the content of section 3 is somewhat out of place; it would be better to move it to the introduction or conclusion as a future research plan.

Section 4 is a description of the authors' proposed ideas, which is somewhat illogical. Figure 1 is the structure of the proposed model, and the subsections should be divided based on Figure 1. All of them should be reflected in the revision.

In particular, Figure 2 describes a dual CNN, but the content of the figure is not clear.  

Section 5 should be reorganized by dividing it into subsections.  

Concerning the numerical analysis of Section 5, I would expect a detailed description of the approach used to system model from Section 4.

Summery: 

1. For the better understanding of readers, the author should be included the illustration of the proposed idea with auther(s)'s main idea. Figure 1 is not enough.

2. The analysis and evaluation of the works seems to be practically and sufficient done. But, this paper needs comparisons with more previous works.

The overall structure and flow of the paper is well organized, but I do not recommend the material for publication due to gaps in the experiment design and lack of novel contribution. 

Relating to the experiment design, the paper is missing much detail relating to what is being measured and how this related to the intended use. 

My final decision is that this work is generally well written paper, but is definitely needed to revise again for publication.

No comments

Reviewer 3 Report

sensors-2473667-peer-review-v2

Comments on sensors-2473667-peer-review-v2:Enhanced Dual Convolutional Neural Network Model Using Explainable Artificial Intelligence of Fault Prioritization for Industrial 4.0.

In this manuscriptthe author proposed the model dataset (DS) with the Adam (AD) optimizer, Ridge Regression (RR) and Feature Mapping (FP). The proposed algorithm had coined with an appropriate acronym DSADRRFP and the same proposed approach aims to leverage the benefits of each component to enhance the overall performance and precision of the predictive model. This ensured the model is up to date and accurate. I really enjoy the process of reading the manuscript, however following comments should be taken into account:

1. In Section 1, the manuscript introduced the background knowledge of the neural network.

However, in this section, the third paragraph is a summary of the goals to be achieved by the proposed algorithm rather than the research completed by the author. So the position of this paragraph needs to be changed.

2. Merging the second and third sections may be a good choice.

3. In Section4, after introducing each step of the algorithm, there is a lack of overall description of Figure 2. Meanwhile, all formulas should be considered for relayout to make them all centered.

4.In Section5, it gives out the verification of the algorithm proposed in the manuscript, and it can easily make the conclusion the algorithm meet the expected requirements.

But the width of all tables should be uniform, the width of the tables in this section is uneven. The serial numbers of all formulas should be aligned to the right, there is a significant deviation in the serial numbers of formula 24. All formulas should be centered rather than left aligned.

5. In Section6, add prospects for the practical use of algorithms may be a good idea. Also, there is no need to summarize the Section1 again. It may be better to directly summarize the algorithm proposed in the article.

In conclusion, this manuscript had given a new method in the field of fault prioritization that used neural network. And the quality of this manuscript needs to be improved.

Round 2

Reviewer 3 Report

The content is generally complete, but the format of some tables does not look perfect, and it is suggested that you can fine-tune it.

The English level of this paper is generally high, but it can still be further improved to make the sentences more beautiful.